# Study protocol: an open-label individually randomised controlled trial to assess the efficacy of artemether-lumefantrine prophylaxis for malaria among forest goers in Cambodia

Richard James Maude [1,2,3,4] Rupam Tripura,[1,2] Mom Ean,[1] Meas Sokha,[1] Thomas Julian Peto [1,2] James John Callery [1,2] Mallika Imwong,[1,5] Ranitha Vongpromek,[1,6] Joel Tarning,[1,2] Mavuto Mukaka,[1,2] Naomi Waithira,[1,2] Oung Soviet,[7] Lorenz von Seidlein [1,2] Siv Sovannaroth[8]

For numbered affiliations see end of article.

**Correspondence to**
Professor Richard James Maude; richard@tropmedres.ac

## ABSTRACT

**Introduction** In the Greater Mekong Subregion, adults are at highest risk for malaria. The most relevant disease vectors bite during daytime and outdoors which makes forest work a high-risk activity for malaria. The absence of effective vector control strategies and limited periods of exposure during forest visits suggest that chemoprophylaxis could be an appropriate strategy to protect forest goers against malaria.

**Methods and analysis** The protocol describes an open-label randomised controlled trial of artemether-lumefantrine (AL) versus multivitamin as prophylaxis against malaria among forest goers aged 16–65 years in rural northeast Cambodia. The primary objective is to compare the efficacy of the artemisinin combination therapy AL versus a multivitamin preparation as defined by the 28-day PCR parasite positivity rate and incidence of confirmed clinical malaria of any species. The sample size is 2200 patient-episodes of duration 1 month in each arm. The duration of follow-up and prophylaxis for each participant is 1, 2 or 3 consecutive 28-day periods, followed by a further 28 days of post-exposure prophylaxis, depending on whether they continue to visit the forest. Analysis will be done both by intention to treat and per protocol.

**Ethics and dissemination** All participants will provide written, informed consent. Ethical approval was obtained from the Oxford Tropical Research Ethics Committee and the Cambodia National Ethics Committee for Health Research. Results will be disseminated by peer-reviewed open access publication together with open data.

**Trial registration number** NCT04041973; Pre-result.

## Strengths and limitations of this study

► Malaria is a major health problem for forest workers in Cambodia. Preventing malaria will provide major health as well as socioeconomic benefits for participants in the first instance and, if rolled out, for forest goers more broadly.

► The trial intervention was designed together with the Cambodian government to be potentially implementable depending on the results.

► Broad engagement with healthcare workers and communities in the study area preceded enrolment and this is continuing throughout with local healthcare workers and forest goers assisting with running the trial including identifying potential participants and supporting follow-up.

► The trial is open label so participants can know which study drug they are taking.

► The outcomes are dependent on the incidence of malaria during the trial follow-up period.

## INTRODUCTION

In the Greater Mekong Subregion (GMS), adults are at highest risk for malaria. The most relevant disease vectors bite during daytime and outdoors which makes forest work a high-risk activity for malaria. The absence of effective vector control strategies and limited periods of exposure during forest visits suggest that chemoprophylaxis could be an appropriate strategy to protect forest goers against malaria.

In the GMS, a large proportion of malaria transmission occurs in forested areas, which serve as perpetual sources of transmission.[1–5] Studies have demonstrated increased risk of malaria among forest goers, particularly in men of working age[6 7] although these have largely been restricted to small geographical areas. Protecting forest goers from Plasmodium infections would not only benefit them directly but also people residing around their home. Malaria elimination efforts which do not consider the reinfection risk from forest

workers are unlikely to succeed. However, preventing infections in forest workers is a major challenge. The biting rhythm and resting behaviour of *Anopheles dirus* reduces the impact of the two most commonly employed control measures, long-lasting insecticide treated bednets (LLIN) and indoor residual spraying. Several studies have also demonstrated poor use of personal protection measures against malaria transmission.[8-10] Two factors that increase malaria risk among forest workers are the basic character of overnight forest accommodation[9] and exposure to the Anopheles vectors (eg, *An. dirus*), which tend to bite outdoors in daytime. LLIN have a high protective efficacy against nocturnal, indoor malaria transmission[11] but are less protective against daytime, outdoor-biting vectors like *An. dirus.* The improvised housing of forest workers is frequently poorly suited to hanging bed nets.[12] Imaginative interventions such as supplying forest workers with insecticide treated hammocks do not address the biting rhythm and resting behaviour of the vectors and have a disappointing uptake in field studies for a variety of reasons including incompatibility with traditional sleep arrangements at home or in the forest.[10 12] In the absence of simple, effective and affordable vector control interventions, providing forest goers with effective antimalarial prophylaxis seems a promising alternative approach to protect them against malaria provided people can be persuaded to take it.[13]

We propose to evaluate the feasibility and protective efficacy of antimalarial prophylaxis during forest work. It has been demonstrated in sub-Saharan Africa that seasonal malaria chemoprophylaxis (SMC) of children, the highest risk group for malaria in tropical Africa, can reduce malaria cases by 75%, is cost effective and safe and can be given by community health workers. [14 15] We propose to provide chemoprophylaxis to forest workers, the population group with the highest malaria risk in the GMS. In the proposed study, we compare chemoprophylaxis with an antimalarial drug, artemether-lumefantrine (AL) to a control agent, multivitamins. A recent mass drug administration in Cambodia demonstrated that dihydroartemisinin (DHA)/piperaquine remains effective to clear low-density, subclinical *Plasmodium falciparum* infections, but there are increasing treatment failures of clinical malaria cases[16] and markers of resistance to piperaquine in Cambodia are increasing. Although artesunate–pyronaridine has recently been introduced for treatment in parts of Cambodia, there remain some unresolved concerns about potential liver toxicity. [17] Evidence to date suggests that efficacy of AL remains high in Cambodia and it is very well tolerated with an excellent toxicity profile and is thus the preferred potential option for prophylaxis by the National Malaria Control Programme (NMCP). However, it must be taken with fat to maximise absorption. Previously it has been difficult or impossible to detect very low-density Plasmodium infections. It is important to do so as low density and asymptomatic infections are an important source of malaria

transmission in Southeast Asia. [18] The availability of more sensitive PCR methods allows us to detect Plasmodium infections with much lower densities. [19 20] By use of PCR, we will be able to detect a difference in the prevalence of low density, subclinical Plasmodium infections between the two study arms in a relatively small sample of study participants and will seek to identify all species of human malaria including *P. falciparum*, *P. vivax* and *P. knowlesi.*

Chemoprophylaxis of forest workers could protect this high-risk group and could reduce or even interrupt transmission in villages. The highly encouraging results of SMC in selected regions of sub-Saharan Africa provide hope that targeting another high-risk group, forest workers, could reduce malaria transmission in Cambodia and the wider GMS. In sub-Saharan Africa, children remain the main risk group for Plasmodium infections. In Southeast Asia, the main risk group are adults working and sleeping outdoors hence we propose to provide chemoprophylaxis for these adults. A major challenge for this strategy is the choice of an appropriate chemoprophylactic regimen in the GMS. The chemoprophylactic regimen of choice in Africa is sulfadoxine/pyrimethamine (S/P) plus amodiaquine despite high level resistance against the S/P component of the regimen. Similarly, we propose the use of AL, a drug whose efficacy remains high in the GMS, unlike, for example DHA/piperaquine. [21] The proposed study will help to assess the efficacy and feasibility of prophylaxis to prevent malaria in forest goers, help to identify the optimal regimen and predict its efficacy in reducing overall transmission. The proposed study is a critical step for future use of chemoprophylaxis to protect forest workers in the GMS against malaria.

## Proposed activities
### AL prophylaxis trial
The study of AL versus a multivitamin preparation will be a two-arm randomised open-label comparative study. Laboratory assessments of malaria infection at baseline and day 28 post forest will be performed blind to treatment allocation and clinical cases during follow-up will be recorded.

### Activities/outcomes
The main activity proposed is an in vivo clinical assessment of prophylaxis to prevent malaria in 4400 participant episodes in 50 villages in Stung Treng Province, Cambodia. The subjects will be randomised in a one-to-one ratio between the artemisinin combination therapy (ACT) AL and a multivitamin preparation with no antimalarial activity.

The study site has been chosen based on current information on incidence of malaria, known predominance of malaria among forest goers, presence of an established clinical research programme and feasibility to perform the proposed research activities.

Efficacy of AL ACT will be assessed through follow-up visits 28 days after returning from the forest on completing each course of prophylaxis when temperature, symptom

questionnaires, brief physical examinations and malaria parasite PCR, and, in selected individuals, parasite genetics will be performed. Episodes of confirmed clinical malaria among study participants at any time point between enrolment and follow-up will also be recorded.

All the organisations in this collaboration will work closely with local counterparts including the NMCP, nongovernmental and other relevant organisations. Training is an integral part of this collaborative working relationship, and the building of local research capacity is an essential component of all research plans.

All research-related activities, from study design, planning, implementation through to analysis and writing of reports will be performed jointly with local counterparts. Both on-the-job training and formal training will be provided when needed, in particular for good clinical practice (GCP) skills.

The close interaction between WHO and its regional offices will ensure that new knowledge is disseminated efficiently and effectively throughout the region.

## METHODS AND ANALYSIS
### Objectives
#### Primary objective
To compare the efficacy of the ACT AL versus a multivitamin preparation as defined by the 28-day PCR parasite positivity rate and incidence of confirmed clinical malaria of any species.

#### Secondary objectives
1. To compare the efficacy of the ACT AL versus a multivitamin preparation as defined by the 28-day, 56-day and 84-day PCR parasite positivity rate and incidence of confirmed clinical malaria for each species.
2. To quantify the impact of the ACT AL as prophylaxis for forest goers on overall malaria transmission using mathematical modelling.
3. To assess the impact of AL prophylaxis on the spread of genetic markers of artemisinin (such as *Kelch13* mutations) and partner drug resistance.
4. To obtain data on the place of residence, work, recent travel history and risk behaviours of forest goers in order to improve the understanding of high-risk groups, locations of malaria transmission and possible routes spread of malaria and artemisinin resistance.
5. To explore the duration, location and purpose of individual forest visits.
6. To obtain detailed data and Global Positioning System (GPS) mapping on a subset of participants and their peers relating to the behaviours and risk factors associated with malaria infection in order to improve understanding of local malaria transmission among forest goers.
7. To determine the prevalence of asymptomatic Plasmodium infections in high-risk populations at varying seasonal time points.

8. To determine the prevalence of other infectious diseases that affect the study population.

### Trial design
#### Study sites
The study will take place at up to 50 villages in selected malaria endemic districts in Stung Treng Province, Cambodia. As the malaria situation in this area is dynamic, the villages will be identified prior to the start of the trial from analysis of up to date malaria incidence from passive surveillance collected by the Cambodia National Center for Parasitology, Entomology and Malaria Control (CNM). The rationale for choosing these areas include high forest cover and ongoing malaria transmission among forest goers. Malaria transmission in this area is generally low but varying over time.

#### Summary of trial design
An open-label randomised parallel group superiority trial among forest goers comparing the ACT AL with a multivitamin with no antimalarial activity to evaluate the efficacy of prophylaxis, and to better understand high-risk groups and locations of malaria transmission. Follow-up will be for 1–3 consecutive periods of 28 days depending on whether the participant continues to visit the forest.

#### Study duration
The recruitment phase of the study is expected to last 12 months. Training and community sensitisation will precede study execution for 3 months. Data management and analysis, sample analysis (PCR, parasite genetics), mathematical modelling and report writing are expected to take about 5 months. The total time to complete the study will be about 20 months.

### Primary and secondary endpoints
#### Composite primary endpoint
1. 28-day PCR positivity rate* of Plasmodium infections of any species and/or
2. Proportion of participants with confirmed clinical malaria of any species reported between day 0 and day 28.

#### Secondary endpoints
1. 28-day, 56-day and 84-day PCR Plasmodium positivity rate for each Plasmodium species.
2. Proportion of participants with confirmed malaria reported between day 0 and day 28 for each species.
3. Description of epidemiological situation of malaria in the study areas from passive surveillance data.**
4. Prevalence of *Kelch13* mutations and other genetic markers of antimalarial drug resistance of known functional significance.
5. Incidence of adverse events (AEs) and serious adverse events (SAEs) by study arm during the course of prophylaxis.
6. Data on the place of residence, work, recent travel history and mobile phone use.

7. Detailed data and GPS mapping on a subset of participants and their peers relating to the behaviours and risk factors associated with malaria infection.
8. Overall prevalence of Plasmodium at baseline, stratified by season and risk factors.
9. Day 0, 28, 56 and 84 capillary blood levels of lumefantrine.
10. Prevalence of serological diagnostic markers of other infectious diseases.

\*PCR positivity rate as determined from the proportion of blood samples that were PCR positive.

\*\*This will include the number of cases per village and demographics of those cases.

## Trial participants
### Overall description of trial participants
Male and non-pregnant female participants aged between 16 years and 65 years planning to visit the forest within 72 hours are the target study population. The upper age limit was chosen as people over 65 years in the study area rarely travel to the forest and are at low risk of malaria. All pregnant women will be excluded as a conservative measure to minimise risk because of insufficient evidence for safety of AL in the first trimester together with frequent uncertainty about the stage of pregnancy, as well as lack of consensus about the required dose in pregnancy. All study participants must meet the applicable inclusion and exclusion criteria.

### Inclusion criteria
1. Male or female, adults aged between 16 and 65 years.
2. Planning to travel to the forest within the next 72 hours and stay overnight.
3. Written informed consent.
4. Willingness and ability of the participants to comply with the study protocol for the duration of the study.

### Exclusion criteria
1. For females: known pregnancy or breast feeding.
2. Participants who have received artemisinin or a derivative or an ACT within the previous 7 days.
3. History of allergy or known contraindication to artemisinins, lumefantrine or multivitamins.
4. Documented or claimed history of cardiac conduction problems.
5. Severe vomiting or diarrhoea on the day of screening.
6. Signs/symptoms of clinical malaria (febrile or history of fever in the previous 24 hours) confirmed by rapid diagnostic test (RDT).

## Procedures
Study procedures will be performed according to the schedule of assessments (online supplemental file 1). This will require that participants are followed up every 28 days for up to three periods on completion of a course of prophylaxis. Enrolment will be done by trained trial staff.

## Informed consent
Prior to the start of enrolment we will conduct community mobilisation and sensitisation activities in each village community where the trial will recruit participants. During the trial, the participant (or witness if illiterate) must personally sign and date the latest approved version of the informed consent form before any study specific procedures are performed. Written and verbal versions of the participant information and informed consent in the local language will be presented to the participants by trained study staff detailing no less than: the exact nature of the study; the implications and constraints of the protocol; and the known side effects and any risks involved in taking part. It will be clearly stated that participation is voluntary and that the participant is free to withdraw from the study at any time for any reason without prejudice to future care, and with no obligation to give the reason for withdrawal.

The participant will be allowed as much time as possible to consider the information and take the opportunity to question the investigator, or other independent parties to decide whether they will (or allow his/her charge to) participate in the study. Written informed consent will then be obtained by means of participant dated signature or thumb print (if unable to write) and dated signature of the person who presented and obtained the informed consent. Examples of the patient information sheet and consent form for this study are provided in English in online supplemental files 2 and 3, respectively.

A copy of the signed informed consent document(s) will be given to the participants.

Children aged 16 to <18 years will be required to sign the latest approved version of the written informed assent form in addition to their parent or guardian signing a consent form.

## Screening, eligibility and baseline assessments
Participants who present at the participating sites will be screened to assess eligibility. Full consent will be obtained before any enrolment procedures are conducted. It will be made clear from the outset that refusal to participate will not jeopardise subsequent antimalarial treatment (if applicable). A screening log will be kept. As detailed below, participants may be enrolled a maximum of three times during the study period. People that cannot return for follow-up as per the schedule will not be enrolled. Known pregnancy will be identified by self reporting.

### Demographics and medical history
Basic demographic and epidemiological data (e.g. sex, age, weight, address, bed net use, malaria risk factors, travel history, prior malaria episodes, prior treatment and previous participation in this or previous studies), and a full medical history will be recorded by the study staff.

### Physical examination and vital signs
Brief physical examination and vital sign will be conducted by a qualified study team member. Weight and

temperature will be documented. A symptom questionnaire will be performed.

### Drug history

All prescribed or over-the-counter and traditional antimalarial medications used within the last 7 days will be recorded. Any drug allergies will be recorded.

### Clinical malaria

Participants who are screened and are found to be febrile or have a current history of fever will not be enrolled (as per exclusion criteria) but will be tested for malaria and, if positive, given antimalarial treatment by the village malaria worker or local clinic. All this will be done in accordance with the current national malaria treatment guidelines in Cambodia. Individuals treated for malaria in this way will not be enrolled in the study as per the exclusion criteria. Such individuals may be enrolled later following recovery provided they meet the inclusion and exclusion criteria.

### Randomisation, allocation and blinding

Participants who fulfil all the inclusion criteria and have none of the exclusion criteria will be randomised 1:1 to one of the two treatment arms according to a randomisation schedule. Randomisation will be in permuted blocks of size that will be determined by the trial statistician and the block size will not be revealed to the investigating team. Randomisation will be stratified by village and villages combined for the analysis. Allocation will be done by trained study staff drawing the next sequential numbered opaque envelope (or other equally reliable randomisation administration procedure), which contains the study number and treatment allocation.

The participants will be assigned a study arm through a computer-generated randomisation schedule. Individual, sealed and sequentially numbered envelopes will be provided for each trial site with one envelope per participant, indicating the treatment allocation.

This is an open-label study so the blinding of investigators and participants is not applicable. However, the randomisation procedure allows for adequate drug allocation concealment before envelopes are opened. All laboratory investigations will be performed without knowledge of the treatment allocation.

### Blood sampling on study enrolment

On study enrolment, immediately before drug administration, blood will be collected for the following:
1. Parasite PCR (up to 1 mL).
2. Storage for later identification of other causes of fever (2 mL).

In case of difficulties with venipuncture on enrolment (eg, due to dehydration, suitably qualified staff not available in the village) or loss of cold chain during transport from remote villages, three dried blood spots will be collected on enrolment for PCR and the other sample collected at follow-up.

### Study drug administration

| Overview of drug regimens | |
|---|---|
| **ACT arm** | **Multivitamin arm** |
| **Artemether-lumefantrine × 3 days followed by 1 day per week, two times per day** | **Multivitamin × 3 days followed by 1 day per week, once daily** |

Participants will be treated with weight-based doses according to the schedule in online supplemental file 1.

The study drugs will be administered by trained study staff.

If the participant vomits within half an hour after intake of the antimalarial drugs, the dose will be repeated. If vomiting occurs between half and 1 hour, half of the dose will be repeated. If vomiting occurs more than 1 hour after drug administration, no repeat dosing will be done. Repeat doses will be recorded on the case report forms (CRFs). If vomiting within 1 hour occurs more than one time, no repeat dosing is allowed. The participant will then be treated at the discretion of the investigator.

The prophylaxis will start with a 3-day course of two times per day AL. This will be followed by two doses 8 hours apart on 1 day per week during the time that the person is travelling in the forest and for 4 weeks after leaving the forest.

### Follow-up

Participants will be asked to return for a follow-up assessment any time from 28 to 35 days after commencing prophylaxis. Twenty-eight days was chosen as the upper limit of the time from infection to detectable parasites in the blood. Ongoing studies in the area found the duration of forest visits varied from a day to several weeks, with very few people being away for more than 28 days. This will be regardless of the duration of their visit to the forest or the number of times they visit it in that period. At this assessment, they will be interviewed about how long they spent in the forest, where they went, why, who they travelled with and about risk factors for infection. Brief physical examination, vital signs and symptom questionnaire will be performed. They will also be asked to report any diagnostic tests and/or treatment for malaria during the preceding 28–35 days.

### Blood sampling at follow-up

At each follow-up visit, the following blood will taken:
*All individuals*:
► Parasite PCR (up to 1 mL).
In those from whom sufficient blood could not be collected at baseline:
► Storage for later identification of other causes of fever (2 mL).
From minimum 100 randomly selected individuals:
► Lumefantrine level (0.2 mL).
*In those with confirmed clinical malaria at any time point between enrolment and follow-up*:

► Dry blood blots (0.4 mL, 3 spots) collected on filter papers for:
  Parasite PCR and DNA genotyping for genetic markers of antimalarial resistance.
  Parasite whole genome sequencing and barcoding to identify geographical origin of parasites and compare genotypes to identify persistent infections.

In individuals who are planning to return again to the forest within the following 28 days after the follow-up visit, they will be asked to continue their weekly prophylaxis according to the original treatment allocation on enrolment. They will then be asked to return for a second follow-up visit a further 28–35 days later when the above procedure will be repeated. This will be repeated one more time. If the person cannot be followed up within the scheduled period, for example, because they do not return from the forest in time, then they will be followed up at the first opportunity and this will be recorded in the CRF.

Thus individuals may take prophylaxis continuously for a maximum of three periods of 28–35 days in the forest plus 4 weeks after returning totaling 112 days. The choice of study medication for each individual will follow the initial assignment on enrolment throughout the follow-up period.

In those who do not declare an intention to return to the forest within 28 days at any follow-up visit, no further follow-up visits will be offered at that time but they will be asked to complete 4 weeks of prophylaxis following their last day in the forest as post-exposure prophylaxis.

Individuals who have been enrolled in the study may be enrolled into the study up to two more times during the 12 months study period only if a minimum period of 28 days (4 weeks) has elapsed following their last dose of prophylaxis. Thus they can be enrolled in the study up to three times. If an individual is enrolled again in this way, they will be re-randomised following the same procedure as enrolment. The rationale for this re-enrolment was that malaria transmission and forest travel are seasonal at this location and this allows detection of malaria positive episodes in people who continue to visit the forest throughout the year while minimising the period of follow-up for the majority of people who visit the forest only during a particular season; in addition, it allows a wash-out period between episodes of taking prophylaxis.

### Time windows

The time-window for the follow-up visits is 28–35 days. If a participant does not attend, the study team will try to locate the participant and conduct the necessary examinations and tests.

### Additional visits

Participants presenting to the village malaria worker, mobile malaria worker or clinic with a fever or other symptoms at any time after enrolment that is not a scheduled study follow-up visit will be assessed and treated by the healthcare workers in the local healthcare system as per routine clinical practice in Cambodia.

On enrolment, participants will be encouraged to attend a village malaria worker or government clinic for the assessment of fever or other symptoms and to report this to the study team as soon as possible. Information on these healthcare encounters including malaria test result and treatment will be recorded in the study CRF.

### Clinical malaria during follow-up

Participants who have an episode of confirmed clinical malaria at any time after enrolment up to the last follow-up visit and for 1 month afterwards will have blood taken for parasite genetic analysis. As clinical malaria at follow-up is part of the composite primary endpoint, and the participants and field staff will not be blinded as to study arm, there is potential for bias if, for example, people in the AL arm choose not to attend for a malaria test. However, extensive efforts will made through community engagement and individual counselling to advise participants against this.

### Blood volumes

The blood volumes for the protocol mandated tests are as follows:
1. PCR: up to 1 mL.
2. Lumefantrine level: 0.2 mL.
3. Dried blood spots for parasite genetics: 0.4 mL.
4. Storage for serology at baseline 2 mL.

Maximum blood volumes are presented below for adults for the maximum of three periods (84 days) of follow-up. The maximum blood volume is the total amount taken if the participants returns for follow-up on three consecutive occasions and had all blood samples taken. The maximum blood volume will be approximately 10.2 mL (less than 10% of total blood volume taken over 8 weeks as recommended by WHO- *Bulletin of the World Health Organization 2011:89:46–53*).

Allowing for the possibility that we may need to repeat blood tests, we may add 10.2 mL to these estimated maximum blood volumes.

Blood samples collected from this study will be stored no longer than 10 years using codes assigned by the study team or their designee(s). Access to research samples will be limited using either a locked room or a locked freezer.

### Analysis of blood samples
### Parasite PCR

This is required for the primary study objective. Blood samples will be analysed in the Molecular Tropical Medicine Laboratory, Bangkok, Thailand using PCR to identify which individuals have malaria parasites of any species. It is anticipated that results will be available around 3–6 months after collecting each sample, thus they will not be used to guide antimalarial treatment at the time of testing. The study teams will be informed which samples were positive for malaria and they will follow-up positive participants to conduct a brief clinical assessment. Any

individuals who are symptomatic will be referred to the village malaria worker or clinic for testing and treatment. The laboratory will be blinded to the study arm of the patient.

### Parasite genetic analysis

Blood samples (dried blood spots) for parasite genetic analysis will be obtained and stored from all subjects recruited with subject's consent. In individuals in whom parasites are found by PCR, samples will be processed for parasite genetic analysis. Genetic samples (in the form of dried blood spots or extracted DNA) will be stored (for a maximum of 10 years) at the Molecular Tropical Medicine Laboratory, Bangkok, Thailand. In those with confirmed clinical malaria, parasite genotyping will be performed at the Wellcome Trust Sanger Institute in Hinxton, UK or other suitable laboratory using a set of informative single nucleotide polymorphisms (SNPs) selected from whole genome sequencing. The subject will be asked for consent for this transfer during the initial informed consent process. A material transfer agreement will be in place if required before any samples are shipped. The results of the parasite genotyping will not be reported back to the subjects. This analysis will only be done for those with confirmed clinical malaria as it is anticipated that there will be insufficient genetic material in samples taken from those with asymptomatic infection due to the low parasite burden in these individuals.

### Lumefantrine level

Blood samples for lumefantrine level will be taken at follow-up visits from a minimum of 100 randomly selected participants, where logistically possible, to assess adherence with the study drugs. These will be analysed in the Pharmacology Laboratory at Mahidol-Oxford Tropical Medicine Research Unit (MORU) in Bangkok, Thailand.

### Serology

Among those who specify by written consent, the serology samples will be analysed for diagnostic markers of other infectious diseases.

### Study drug

#### Artemether-lumefantrine

AL is currently available as standard tablets containing 20 mg artemether and 120 mg of lumefantrine in a fixed-dose combination formulation. It is included in this formulation on the WHO Model List of Essential Medicines.[22]

#### *Target dose/range*

The AL is administered as a two times per day dose for 3 days for a total of six doses (an initial dose, second dose after 8 hours and then two times per day—morning and evening—for the following 2 days) followed by two times per day once a week according to the treatment schedule in online supplemental file 1.

### Multivitamin

The multivitamin preparation will be HEXA CMP (Chemephand Medical) or suitable equivalent alternative administered as a once daily dose using the treatment schedule in online supplemental file 1. This multivitamin does not contain any compound with antimalarial activity. Its components are: vitamin A: 5000 USP units, vitamin D: 400 USP Units, ascorbic acid: 75 mg, thiamine mononitrate: 2 mg, riboflavin: 3 mg and niacin amide: 20 mg. A multivitamin was chosen because a placebo was not available from the manufacturer for this trial. The multivitamin has no effect on malaria, is safe, is acceptable to the community and is easily available at the study site. Providing a medication to all participants makes it easier to explain the study in a way that is socially acceptable, and has potential to discourage the sharing of study drugs by participants in the two study arms.

### Storage of study drugs

All efforts will be made to store the study drugs in accordance with the manufacturers' recommendations in a secure area. This may be difficult at some sites where air-conditioned storage rooms are not available. The ACT should be stored between 15°C and 30°C (59°F–86°F).

Where this is not possible and monitored storage conditions do not meet the recommendations, the artemisinin-derivatives and partner drug content of batches of ACT will be retested at the end of the study.

### Compliance with study drugs

Study drugs will be administered as directly-observed-therapy (DOT) on the first day. Where possible, study drugs will also be administered as DOT on days 2 and 3. Where DOT is not possible, the participant will be contacted by the study team by telephone or in person to ensure they take the second and third days of medication and to ensure they follow the correct procedure in case of vomiting. If the participant vomits, and is re-dosed; this will be recorded in the CRF. If vomiting within 1 hour occurs again after retreatment, no repeat dosing is allowed. All drug doses will be recorded in the CRF. To maximise adherence to the study medication, the study will be preceded by a period of community sensitisation and engagement including information sessions on the importance of taking all doses of medication. The participants will be requested to take each dose with food to maximise absorption of the lumefantrine.

### Accountability of the study treatment

All movements of study medication will be recorded. Both study medication of individual participant and overall drug accountability records will be kept up to date by the study staff.

### Concomitant medication

Throughout the study, investigators may prescribe concomitant medications or treatments deemed necessary (eg, antipyretics or anti-emetics) to provide adequate supportive care except for antibiotics with antimalarial

activity unless unavoidable (eg, doxycycline, azithromycin). If these are required, the participants will be kept in the study and this will be noted as a protocol deviation. Anti-emetics should not be prescribed as a prophylaxis if no nausea or vomiting is present.

Antimalarials for symptomatic, confirmed malaria infections will be prescribed as described above. Any medication, other than the study medication taken during the study will be recorded in the CRF.

### Epidemiological data on place of residence, work, travel history and malaria risk

In order to have a greater understanding of the possible sites of malaria transmission, and to relate genetic diversity to geographical location, participants will be asked a short set of questions on their place of residence, place of work and their history of travel plus possible risk factors for malaria. This is to obtain a detailed understanding of the behaviours and risk factors for malaria infection. We will collect GPS coordinates of the places of residence of all participants. In a subset of participants, GPS coordinates will be collected for their travel patterns during follow-up including place of work, forests, forest camps, farms or plantations to identify places where their infection may have occurred. The size of this subset will be determined by the availability of GPS devices with the number being limited to 50 participants at any one time. The GPS devices will be offered to unselected consecutive trial participants whenever they are available, being returned on completion of follow-up for that individual. We will collect all available local malaria treatment records to describe how the study population compares to the overall population who receive treatment for malaria and this will allow us to better understand local malaria epidemiology and transmission patterns. All personal information will be anonymised so that no individual can be identified from their treatment records, through interviews or from mapping data.

### Malaria incidence data

Passive surveillance data from all available sources for the study province collected by CNM will be analysed to identify any changes in malaria incidence rate in study villages before, during and after the study where AL prophylaxis was administered compared with non-study villages.

Enrolled participants who experience an episode of confirmed clinical malaria during follow-up will be linked back to their individual case records to quantify the incidence of clinical malaria in each study arm.

### Analysis
#### PCR for parasites

PCR will be used to identify which individuals have parasites at enrolment (prior to taking the study medicine) and at each follow-up visit and is required for the primary study objective.

#### Parasite genetics

Parasite DNA will be used for genomic studies including but not limited to parasite species confirmation, microsatellite typing to identify parasite clones and SNPs typing/whole genome sequencing to generate data for studies of the geographic origins of the parasites.

#### Lumefantrine levels

Lumefantrine levels will be used to assess adherence in a random sample of study participants.

#### Serology

The serology sample will be used for anonymised investigation of the prevalence, incidence, association with fever, and risk factors for other common infectious diseases affecting the study population. Samples will be stored for later analysis.

### Retention, discontinuation/withdrawal of participants from the study

All efforts will be made to retain as much data as possible. The main strategy that is being re-enforced for data retention includes study staff reminding participants of the upcoming data collection. This will be emphasised during management training. However, each participant has the right to discontinue the study drug or the study at any time. Data accrued up until the time of discontinuation will be used in the analysis.

In general, the investigators will be required to make every effort to perform the study procedures until completion of follow-up (maximum three visits over 84 days), including in the following situations:
► Significant non-compliance with treatment regimen or study requirements.
► An AE which requires discontinuation of the study medication or results in inability to continue to comply with study procedures.
► Disease which requires discontinuation of the study medication or results in inability to continue to comply with study procedures.
► Loss to follow-up (every attempt should be made to re-contact the participant).

However, the investigator may discontinue participation in the study of a participant if he or she considers it necessary.

In addition, the participants always have the right to withdraw consent in writing or verbally.

The reason for withdrawal or discontinuation, if available, will be recorded in the CRF. If the study drug or participation in the study is discontinued due to an AE, the investigator will arrange for follow-up visits at least until the AE has resolved or stabilised.

If a participant does become pregnant during participation in the study, they will be withdrawn from the study immediately on it being reported to the study team. Any pregnancy must be reported to the principal investigator (PI) within one working day of awareness. The PI must take all reasonable efforts to discover the outcome of the

pregnancy and fill out the pregnancy form. If there is a congenital abnormality or a stillborn baby, this needs to be reported as an SAE.

## Source data

Source documents are original documents, data, and records from which participants' CRF data are obtained. These include, but are not limited to, village malaria and clinic records (from which medical history and previous and concomitant medication may be summarised into the CRF), clinical and office charts, laboratory and pharmacy records, diaries, microfiches, radiographs and CRFs.

CRF entries will be considered source data if the CRF is the site of the original recording (e.g. there is no other written or electronic record of data). In this study, the CRF will be used as the source document for most of the data points.

All documents will be stored safely in confidential conditions.

## Safety reporting

This trial will use drugs that have either been registered or evaluated extensively. To add to the evidence base for safety of AL as prophylaxis, we will record and review all AEs and SAEs that are reported to occur in the study.

A symptom questionnaire will be performed on enrolment and at each subsequent follow-up visit to the healthcare centre, to aid in the identification of AEs. In addition, enrolled individuals will be encouraged to promptly report any unexpected symptoms or illnesses between follow-up visits to the study team.

The investigator is responsible for the detection and documentation of events meeting the criteria and definition of an AE or SAE, as provided in this protocol.

All SAEs and AEs will be promptly documented from the moment of drug administration in the study to discontinuation of the participant from study participation. Any events occurring between screening and drug administration will be considered as baseline, preexisting conditions.

All AEs must be recorded in the AE/SAE CRF. To avoid colloquial expressions, the AE should be reported in standard medical terminology. Whenever possible, the AE should be evaluated and reported as a diagnosis rather than as individual signs or symptoms. If a definitive diagnosis is not possible, the individual symptoms and signs should be recorded. Whenever possible, the aetiology of the abnormal findings will be documented on the CRF. Any additional relevant laboratory results obtained by the investigator during the course of this study will be recorded on the CRF.

If the event meets the criteria for 'serious', the SAE must be reported to the PAL-Cambodia safety team within 24 hours of the time that the event was identified. If further data are required, additional documentation can be submitted. All SAEs must be followed until resolution, or until the SAE is deemed permanent or leads to death.

Samples will be shipped for PCR to a molecular laboratory where they will be analysed in batches. Quality control results will be available approximately 3–6 months from the time of collection. The list of positive tests will be returned to the field sites. If a participant is found to have a Plasmodium infection, and has not already received antimalarial treatment subsequent to the sample being collected, then these individuals will be contacted by a local health worker, and if a participant reports fever or illness they will be offered appropriate diagnosis and treatment.

## Definitions

### Adverse event

An AE is any undesirable event or clinical deterioration that occurs to a study participant during the course of the study; that is, from the time of administration of study drugs until study ends (i.e. until the follow-up visit) whether or not that event is considered related to the study drugs, or to a concomitant drug or procedure: for example,

1. Any unfavourable and unintended symptom.
2. Physical sign.
3. Abnormal laboratory result.
4. An illness.

Any new clinical sign or clinical deterioration that occurs between signing the consent form and the administration of study drugs is not an AE. This information will be recorded in the medical records, as a pre-existing condition.

### Serious adverse event

An SAE is an AE that:

► Results in death.
► Is life-threatening that is, the participant was at risk of death at the time of the AE.
► Requires in participant hospitalisation or prolongation of existing hospitalisation.
► Results in persistent or significant disability/incapacity.
► Is a congenital anomaly/birth defect.
► Any other significant medical condition.

All of the above criteria apply to the case as a whole and should not be confused with the outcomes of individual reactions/events. More than one of the above criteria can be applicable to the one event. Important medical events that may not be immediately life-threatening or result in death or hospitalisation may be considered an SAE when, based on appropriate medical judgement, they may jeopardise the participant or require medical or surgical intervention to prevent one of the outcomes listed in the definition above. Examples of such medical events include allergic bronchospasm requiring intensive treatment in an emergency room or at home, blood dyscrasias or convulsions that do not result in hospitalisation, or development of drug dependency or drug abuse.

### Reporting procedures for SAEs

All SAEs must be reported by the site investigator to the Study PI and PAL-Cambodia safety and medical monitor, within 1 day of his or her awareness of the SAE. The SAE report, should be emailed to the email paltrial@tropmedres.ac.

Further reports should be submitted, if required, until the SAE is resolved.

The site investigator must also report the SAEs to the local ethics committee in accordance with local requirements.

### Evaluating AEs and SAEs
#### Assessment of intensity

Each AE will be graded according to the Common Terminology Criteria for Adverse Events (CTCAE) V.5.0 November 2017.

If an AE is not listed in the CTCAE table, the investigator will assess the severity using the following guidelines:

1=Mild; asymptomatic or mild symptoms; clinical or diagnostic observations only; intervention not indicated.

2=Moderate; minimal, local or noninvasive intervention indicated; limiting age appropriate instrumental ADL.*

3=Severe or medically significant but not immediately life-threatening; hospitalisation or prolongation of hospitalisation indicated; disabling; limiting self care ADL**

4=Life-threatening consequences; urgent intervention indicated.

5=Death related to AE.

Activities of Daily Living (ADL).

*Instrumental ADL refer to preparing meals, shopping for groceries or clothes, using the telephone, managing money and so on.

**Self care ADL refer to bathing, dressing and undressing, feeding self, using the toilet, taking medications and not bedridden.

### Clarification of the difference in meaning between 'severe' and 'serious'

The term 'severe' is often used to describe the intensity (severity) of a specific event (as in mild, moderate or severe myocardial infarction); the event itself, however, may be of relatively minor medical significance (such as severe headache). This is not the same as 'serious', which is based on the outcome or criteria defined under the SAE definition. An event can be considered serious without being severe if it conforms to the seriousness criteria, similarly severe events that do not conform to the criteria are not necessarily serious. Seriousness (not severity) serves as a guide for defining regulatory reporting obligations.

### Assessment of relatedness

The investigator is obligated to assess the relationship between study drug and the occurrence of each AE/SAE using the following categories of relatedness:

1. Definite: clear-cut temporal association.
2. Probable: clear-cut temporal association, with improvement on drug withdrawal, and not reasonably explained by the participant's known clinical state or other aetiology.
3. Possible: less clear temporal association; other aetiologies are possible (other possible aetiologies should be recorded on the CRF).
4. Not related: no temporal association with the study drug; assessed as related to other aetiologies such as concomitant medications or conditions, or participant's known clinical state.

The investigator will provide the assessment of causality as per the AE/SAE data collection tool.

### Outcome

The investigator will follow-up the AE and SAE until resolution or until no further medically relevant information can be expected. AE and SAE outcome will be classified as follows:

1. Continuing/ongoing.
2. Resolved.
3. Resolved with sequelae.
4. Permanent.
5. Fatal.

### Statistical considerations
#### Sample size justification

The target population for this study will be adult Cambodians who work and sleep in the forest (farmers, collect forest goods, hunting and so on). Two thousand two hundred study participant episodes are required in each arm to have sufficient power to detect a statistically significant difference between the treatment arm and a control arm. An episode is defined as a follow-up period of 28 days with each enrolled individual contributing 1, 2 or 3 episodes. The estimate of the required sample size is complicated by the scarce data on *P. falciparum* incidence in forest workers.

Formally, we anticipate that the risk of being *P. falciparum* positive without receiving prophylaxis will be around 5%. A total of 1605 participant-episodes per arm are enough to detect a difference of at least 40% in the proportion of episodes with a *P. falciparum* positive result as defined by the 28-day PCR parasite positivity rate, that is from 5% positivity in participants without receiving antimalarial prophylaxis (ie, multivitamin) to 3% positivity in participants receiving AL prophylaxis. This has been estimated with 80% power and 5% significance level. However, we also anticipate that we will likely observe multiple episodes being recruited into the study that can reduce power of the study if not accounted for. To compensate for the multiple episodes and any losses to follow-up, we plan to recruit approximately 600 (ie, 595) additional episodes in each group on top of the required 1605 single episodes. This gives an additional 27% episodes to account for the multiple episodes and losses to follow-up. Thus, the overall sample size will be 4400 episodes (ie, 2200 episodes in the treatment arm and 2200 episodes in the control arm). The sample size calculations have been performed in Stata V.15.

### Statistical analyses

The main analysis strategy for the primary outcome will be the intention to treat (ITT) principle followed by the

per protocol (PP) analysis. Thus, we will first analyse the ITT population in which all participants recruited in the trial will be included in the analysis according to the randomisation arm irrespective of what they actually got. These ITT analyses will be followed by the analysis of the PP population in which participants who did not adhere to the protocol will be excluded from analysis.

The composite primary endpoint will be analysed as follows. For 28-day PCR Plasmodium positivity and parasite positive clinical episode rate analysis, each arm will be summarised using crude proportions and binomial exact 95% CIs. The risk differences in Plasmodium positivity between AL versus multivitamin will be reported along with the corresponding 95% CIs. Robust standard errors will be used to handle multiple episodes. These analyses will be complemented by the use of the crude Kaplan-Meier estimates of cumulative PCR Plasmodium positivity and parasite positive clinical episode probabilities as recommended by WHO. The incidence of confirmed clinical malaria between day 0 and day 28 analysis will be modelled using the mixed effects Poisson regression model to obtain incidence rate ratios comparing AL versus multivitamin arms. The mixed effects models will take into account the correlation of multiple episodes from the same participant. Tests of significance will be performed at 5% significance level. Analysis of all endpoints will be described in detail in a Statistical Analysis Plan finalised prior to locking the database. A brief overview is given below.

### Proportions
These will be compared using chi squared or Fisher's exact test, as appropriate. Crude proportions will be calculated with the exact 95% CIs, where relevant.

### Continuous data
These will be summarised by medians (IQR, ranges) and means (SD, 95% CIs), as appropriate, and will include the parasite counts and laboratory parameters. Comparisons of continuous data will be assessed using the paired/unpaired t-tests or the signed rank/Mann-Whitney U tests, as appropriate.

### Safety analysis
Safety analyses will be based on the whole population that get administered the study drug. Safety and tolerability of ACT versus multivitamin will be assessed by comparing the frequency (%) of AEs and SAEs, with particular attention to abdominal pain and appetite perturbation, using the Fisher's exact test. Safety data will be presented in tabular and/or graphical format and summarised descriptively. Any clinically relevant abnormalities or values of potential clinically concern will be described. Participants will be analysed according to an ITT and a PP method where appropriate.

### Handling of missing data
For analyses of proportions, missing outcomes will be imputed using plausible values. For example, worst-case scenario may be deemed appropriate and in that case

sensitivity analysis will be performed with the best-case scenario. In the ITT Kaplan-Meier/survival analysis, participants who are lost to follow-up, or who have Plasmodium reinfections or inconclusive PCR correction, will be censored from the moment of occurrence of one of these events. This survival analysis approach is the best way of handling missing data because participants with partial information are included in the analysis up to the time when they are lost/withdraw from the study.

### Adverse events
AEs will be graded according to CTCAE V.5.0, November 2017.

All AEs that are newly started or increased in intensity after the study drug administration will be reported. AE reports will be generated for all AEs that occurred after study drug administration, until the end of the study.

### Mathematical modelling
The impact of the ACT AL as prophylaxis for forest goers on overall malaria transmission will be quantified using mathematical modelling. For this we will develop a population dynamic village level individual-based model of malaria transmission and treatment parameterised with published data, results from the analysis of data from the trial and fitted to surveillance data from the study area.

### Direct access to source documents/data
Direct access will be granted to authorised representatives from the sponsor and host institution, the regulatory authorities, and ethical committee (if applicable), to permit trial-related monitoring and inspections.

### Quality control and quality assurance procedures
The study will be conducted in accordance with the current approved protocol, International Conference on Harmonization (ICH) GCP, any national regulations that may apply to this study and standard operating procedures. The WorldWide Antimalarial Resistance Network (WWARN) will be engaged in assuring quality assurance (QA)/quality control (QC) of study execution in collaboration with the MORU Clinical Trials Support Group (CTSG). Their role will include but not be limited to monitoring adherence to standard operating procedures (SOPs) for collection of laboratory specimens and quality checks (curation) of laboratory data according to standard methodologies.

### Monitoring
Study sites may have in place a system for internal monitoring. In addition, regular external monitoring of all sites will be performed by the MORU CTSG according to ICH GCP and a Monitoring Plan. Data will be evaluated for compliance with the protocol and accuracy in relation to source documents. The monitors will check whether the clinical trial is conducted and data are generated, documented and reported in compliance with the protocol, GCP and the applicable regulatory requirements. Evaluation of on-site monitoring schemes, such as

a reciprocal monitoring scheme, may be undertaken at selected sites by CTSG.

## Patient and public involvement

During 2018, extensive consultations were held with local authorities, patients, and study communities regarding the design and organisation of the trial. This included the district health authorities and Governor's office. The Siem Pang field station conducted malaria treatment studies and as part of these studies interviews and questionnaires were done with patients to better understand the local risk factors for malaria, travel histories and the nature of forest work.[23] Specific questions on the time spent in forests[24] and the use of medicines in forests and the willingness to take antimalarial prophylaxis were asked. A review of forest acquired malaria was prepared at the site in collaboration with local partners.[25] The potential importance of antimalarial prophylaxis was identified through understanding of the high risk of malaria infection in local forests and the willingness of participants to take medicine to prevent this. Through conversations with patients with malaria treated at the health centre and from monthly meetings with village malaria health workers the design of the study was informed. This supported decisions about: time spent in forests, follow-up scheduling, type of sample collection, monitoring of treatment compliance, suitable locations and communities where patients could be enrolled, concerns and questions surrounding AEs, and defining the messaging and rationale in local languages. Recruitment takes place in villages and community leaders and local health workers including village malaria workers are part of the study team. Patients previously enrolled in studies often serve as guides and assistants as they trust the study team and know the local community and as they are often forest workers themselves they know other forest workers like them who may be willing to participate. Conversations were held with patients and local stakeholders regarding feasibility and to ensure that participation in research would be acceptable and not burdensome or interfering with regular activities. From these discussions, we adopted an outreach strategy so that follow-up can take place without participants needing to travel long distances or to the health centres. Dissemination will take place on several levels: in villages, at district level, at provincial level and at a national and international level. As part of an ongoing research platform in the district, we will communicate results back to study communities at the end of the trial by public meetings. A series of public engagement activities, including dissemination activities, has run alongside the CNM-MORU malaria field studies since 2013.[26]

## ETHICS AND DISSEMINATION
### Declaration of Helsinki

The Investigator will ensure that this study is conducted in compliance with the current revision of the Declaration of Helsinki.

### ICH guidelines for GCP

The investigators will ensure that this study is conducted according to any National Regulations and that it will follow the principles of the ICH Guidelines for GCP.

### Approvals

The study protocol and its associated documents will be submitted to the Oxford Tropical Research Ethics Committee and the appropriate local ethics committees for written approval.

The investigator will submit and, where necessary, obtain approval from the above parties for all substantial amendments to the original approved documents.

### Risks

This study will use drugs that have been studied thoroughly and their toxicities are well described. In general, they are all well tolerated. In the event of any serious or severe AE participants will be referred to the local referral hospital where best available care will be provided.

#### Risks of AL

The safety of artemether and lumefantrine for treatment of malaria has been evaluated in clinical trials and, post licensing, widespread use for treating malaria in hundreds of millions of patients per year. Reported AL side effects have generally been mild. Reported adverse reactions in clinical trials have been similar or lower in frequency and magnitude to other ACTs. The most common (>=3%) reported AEs in clinical studies with AL in adults were headache, anorexia, dizziness and asthenia. AL is not known to cause harmful prolongation of the corrected QT interval (QTc).[27]

#### Risks of multivitamin

The main side-effects of multivitamin are upset stomach, unpleasant taste or headache which are mild to moderate in nature. Very rarely, these may cause an allergic reaction.

#### Risk of phlebotomy and finger prick

The primary risks of phlebotomy include local discomfort, occasional bleeding or bruising of the skin at the site of needle puncture, and rarely haematoma or infection. Phlebotomy will be performed by suitably qualified and trained staff using appropriate hygiene measures including gloves and alcohol swabs to clean the skin.

#### Risk of GPS data

Due to the potential unique nature of the GPS tracking data, it may be possible to identify individuals from their tracks. This will be minimised by the GPS tracking data being kept separately from any personally identifiable information and linked to the data collected on the study CRF only through a unique study code. The GPS tracking data will also be stored anonymously on the tracking device during collection and moved to an encrypted hard drive on completion of collection.

## Benefits

There are no anticipated direct benefits to the participants in this study. However, knowledge gained from this study is expected to help to assess the efficacy and feasibility of prophylaxis to prevent malaria in forest workers, and to predict its efficacy in reducing overall transmission. The proposed study is a critical step for future use of chemoprophylaxis to protect forest workers in the GMS against malaria.

## Alternatives to study participation

Participants are able to decline freely participation in this study. If so, they will receive standard care for their malaria (if applicable).

Study participants will be compensated for time lost from work as a result of trial activities, the cost of local transport to attend for the follow-up visits and will receive a per diem to cover the costs of meals on those days. The amounts in monetary terms will be determined by CNM in accordance with local norms.

The study will pay for treatment for drug-related SAEs or other research-related injuries. The study cannot pay for long-term care for disability resulting from complications of the illness.

## Confidentiality

The trial staff will ensure that the participants' anonymity is maintained. The participants will be identified only by initials and a study number on the CRF and electronic databases. All documents will be stored securely and be accessible to trial staff and authorised personnel only.

## Sample sharing and storage

Samples collected will be used for the purpose of this study as stated in the protocol and stored for future use no longer than 10 years. Consent will be obtained from participants for sample storage and/or shipment of specific samples to collaborating institutions for investigations that cannot be performed locally. Any proposed plans to use samples other than for those investigations detailed in this protocol will be submitted to the relevant ethics committees prior to any testing. Material transfer agreements will be arranged and signed where appropriate/needed.

## Data handling and record keeping

Study data will be recorded on the CRF at the study sites and stored in a secure database. Validation checks will be built into the study database to identify missing values, inconsistencies or invalid data. Additionally, study data will be profiled using statistical software to check for outliers and errors not detected by the database. All tasks related to data management will be carried out in accordance with the study data management plan.

## Data sharing

De-identified, individual participant data from this study will be available to researchers whose proposed purpose of use is approved by the data access committee at MORU. Enquiries or requests for the data may be sent to datasharing@tropmedres.ac.

## Sponsorship and insurance

The University of Oxford has a specialist insurance policy in place: Newline Underwriting Management, at Lloyd's of London—which would operate in the event of any participant suffering harm as a result of their involvement in the research.

## Dissemination plan

Results will be published in the open access peer-reviewed medical literature. Any data published will protect the identity of the participants. This trial will be registered in a web based protocol registration scheme. All those who have made a substantial contribution will be coauthors on publications. The sites have the right to publish their data individually and to include members of the sponsor's team who have made a significant contribution. There will also be publications of pooled data which will be coordinated by the MORU group. All sites will have the opportunity to contribute to these publications.

All the research findings from the programme and from relevant research outside the Programme will be analysed and integrated, and through the WHO Global Malaria Programme will be disseminated to policy makers, NMCPs and other researchers.

## Standard Protocol Items: Recommendations for Interventional Trials checklist

A completed Standard Protocol Items: Recommendations for Interventional Trials checklist for the protocol is provided in online supplemental file 4.

**Author affiliations**

[1]Mahidol-Oxford Tropical Medicine Research Unit, Faculty of Tropical Medicine, Mahidol University, Bangkok, Thailand

[2]Centre for Tropical Medicine and Global Health, Nuffield Department of Medicine, University of Oxford, Oxford, UK

[3]Harvard TH Chan School of Public Health, Harvard University, Boston, MA, USA

[4]The Open University, Milton Keynes, UK

[5]Department of Molecular Tropical Medicine and Genetics, Faculty of Tropical Medicine, Mahidol University, Bangkok, Thailand

[6]Asia-Pacific Regional Centre, WorldWide Antimalarial Resistance Network, Bangkok, Thailand

[7]Provincial Health Department, Stung Treng, Cambodia

[8]National Center for Parasitology, Entomology and Malaria Control, Phnom Penh, Cambodia

**Contributors** RJM designed the study, wrote the protocol. RT contributed to overall study design, edited the protocol. ME contributed to design of the field data collection, edited the protocol. SM contributed to design of the field data collection, edited the protocol. TJP contributed to overall study design, edited the protocol. JJC contributed to overall study design, edited the protocol. MI contributed to study design for PCR and genetic analysis, edited the protocol. RV contributed to study design for sample collection, edited the protocol. JT contributed to study design

for the dosing schedule and analysis for lumefantrine levels, edited the protocol. MM contributed to study design for randomisation, sample size calculation and statistical analysis, edited the protocol. NW designed the data management plan, edited the protocol. OS contributed to study design for field data collection, edited the protocol. LvS contributed to overall study design, edited the protocol. SV designed the study, edited the protocol.

**Funding** This work was supported by The Global Fund to Fight AIDS, Tuberculosis and Malaria grant number B9R03070 and the Wellcome Trust of Great Britain grant number 106698/Z/14/Z.

**Disclaimer** The sponsor and funder had no role in the in study design; collection, management, analysis and interpretation of data; writing of the report or the decision to submit the report for publication and they do not have ultimate authority over any of these activities.

**Competing interests** None declared.

**Patient consent for publication** Not required.

**Provenance and peer review** Not commissioned; externally peer reviewed.

**ORCID iDs**
Richard James Maude http://orcid.org/0000-0002-5355-0562
Thomas Julian Peto http://orcid.org/0000-0003-3197-9891
James John Callery http://orcid.org/0000-0002-3218-2166
Lorenz von Seidlein http://orcid.org/0000-0002-0282-6469

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
