## [Reviewer comments · BMJ Open]

ARTICLE DETAILS

TITLE (PROVISIONAL)	Study Protocol: an open-label individually randomised controlled trial to assess the efficacy of artemether-lumefantrine prophylaxis for malaria among forest goers in Cambodia
AUTHORS	Maude, Richard; Tripura, Rupam; Ean, Mom; Meas, Sokha; Peto, Thomas; Callery, James; Imwong, Mallika; Vongpromek, Ranitha; Tarning, Joel; Mukaka, Mavuto; Waithira, Naomi; Soviet, Oung; von Seidlein, Lorenz; Sovannaroth, Siv

VERSION 1 – REVIEW

REVIEWER	Dr Daniel Cooper Cambridge University Hospitals, Cambridge, UK. Menzies School of Health Research, Darwin, Australia
REVIEW RETURNED	16-Nov-2020

GENERAL COMMENTS	The authors present a protocol for a an open-labelled randomised controlled trial to assess the efficacy of artemether-lumefantrine prophylaxis for malaria among forest goers in Cambodia. They present a well-designed and achievable trial that addresses an important public health question in the region. I have no major concerns about the study design or potential of the proposed trial to assess the hypotheses presented. The sample size justification is well considered and based on robust prior studies and data. I have listed some minor points below, mostly for clarification and ease of understanding of the protocol. The SPIRIT guideline points listed at the end should be specifically addressed and included, as per BMJ Open reviewer guidance and SPIRIT guidelines. 1. The authors state that previous interventions with the same aim have had disappointing uptake in field studies (Pg 4, Line 26). There is no discussion about potential adherence (or lack) to prophylaxis. Why would prophylaxis be better adhered to than previous interventions?2. This is an open-label study. Why are multivitamins being used in the "control" arm? Is there rationale for a benefit of multivitamins in malaria prophylaxis? If this is intended as a "placebo", this should be discussed. If not, and there is no equipoise between the two interventions, this should be more explicit. Addendum: The authors state on page 11 that no components included in the multivitamin have antimalarial activity. The reason for choosing this as a comparator arm therefore needs to be discussed in more detail.3. There is no discussion of other Plasmodium species with important transmission within the SouthEast Asian region. In particular, P. knowlesi has key transmission within forest goers and forest dwellers (Grigg et al, Lancet planet Health, 2017) and has been confirmed in clinical cases in Cambodia (Khim et al, EID,
---

2011).

4. The authors discuss the ability to detect low density, sub-clinical infection (inferred as a benefit to aid sample size and analysis) Pg 4, Line 49. Is there evidence of benefit to detecting and/or treating low-density subclinical infection? Accepting that this isn't an aim of the study, this should be discussed in terms of extrapolating clinical/public health benefit from the outcomes.

5. As the primary objective currently reads, it suggests essentially a composite endpoint (PCR positivity OR incidence of confirmed clinical malaria), however the endpoints are listed as co-primary endpoints. It is not currently clear whether this will be analysed as a composite endpoint, or as two separate endpoints. Both are valid, but either way, I think this needs more clarification, and explicit acknowledgement somewhere within the manuscript of a composite endpoint (not necessarily within the main definition) If that is the plan.

6. It is not clear whether PCR positivity is exclusively within asymptomatic participants. This should be more clear.

7. It is not clear to what extent infections/individuals within the two co-primary endpoints will overlap (if not a composite endpoint). This could be acknowledged/discussed.

8. The reason for an upper age limit on inclusion should be discussed.

9. artemether-lumefantrine is considered safe in second and third trimesters of pregnancy, the reason for exclusion of this group should be discussed.

10. Will randomisation be stratified by site?

11. The authors discuss on page 8, lines 53 – 56 that this is an open-labelled trial and therefore blinding will not be achieved to participants or investigators. However, it is worth discussing that one of the co-primary endpoints is entirely objective (PCR positivity) with blinded laboratory investigations, however the diagnosis of clinical malaria may in theory be subject to bias (however unlikely that is).

12. The rationale for choosing follow-up duration is not discussed.

13. The authors have provided a brief statistical analysis plan. They have not, however, stated how the co-primary endpoints will be analysed. This should be stated (see further below).

Specific SPIRIT guideline points to be clarified (please include each of these points within the body of the text as per SPIRIT guidelines):

Allocation implementation (Item 16 c)

The person(s) who will enrol/assign participants

Data collection methods - retention (Item 18 b)

Strategies to promote participant retention and complete follow up (e.g. phone call reminders, financial compensation)

Statistical methods (Item 20 a)

The main analysis of the primary outcome including the analysis methods to be used for statistical comparisons

	The effect measure for the primary outcome (e.g. OR, RR, difference in means) Significance level and/or intended use of confidence intervals Population analysed (Item 20 c) Clear indication of which participants will be included in the main analysis. Simply stating intention-to-treat or per-protocol without further specification is not sufficient. How missing data will be handled (or a description of why missing data is unlikely)
--	---

REVIEWER	Chaturaka Rodrigo UNSW Sydney, Australia
REVIEW RETURNED	24-Nov-2020

GENERAL COMMENTS	I have read this protocol with interest, which is well-written and addresses an important research question. However, I have the following concerns regarding some aspects of the protocol  1. While the randomisation process for individual recruits are explained, these individuals will be selected from particular villeges and the random selection of villages is not explained 2. If this is a open label trial, what is the advantage of giving multivitamin tablets to control group compared to no treatment? 3. One of the objectives refer to a "mathematical modelling" with no details of how this will be done in the protocol 4. There are several objectives (examining high risk behaviours) that may be beyond the scope of the drug efficacy trial protocol (while I agree these are important for prevention of malaria, the purpose of including these details in this protocol is unclear) 5. There is a risk of including patients with subclinical malaria infection at baseline and saving a sample for later PCR does not solve this problem. Alternatively you may restrict your analysis to clinical malaria 6. The procedure for handling missing data is not clear apart from that data collected up to the last point of enrolment will be included for dropouts. Will it be a intention to treat analysis? 7. I can foresee some serious ethical concerns with some types of data collected (GPS data on participants) and this protocol should ideally be approved by an ethics review committee and the editor may decide if it should be a pre-requisite for publication 8. Testing for genetic markers is vague (e.g. Kelch13 and other genetic markers...). Please specify what you are specifically looking for 9. If a patient had clinical malaria at baseline, is he/she eigible to be recruited after treatment and eradication of infection? 10. If the same person is re-enrolled (as mentioned in the protocol), how will this be addressed statistically as it is incorrect to use the same statistical test when some obervations are within host and others are between-host. Several other comments (including the ones listed above) are also given as annotations in the attached PDF The reviewer provided a marked copy with additional comments. Please contact the publisher for full details.
---

REVIEWER	Jeremy Keenan UCSF, USA
REVIEW RETURNED	10-Dec-2020

GENERAL COMMENTS

This is a trial protocol of a randomized trial comparing antimalarial prophylaxis with artemether-lumefantrine vs a multivitamin control among forest-goers in Cambodia. My main comments are that (1) I do not completely understand the rationale for conditioning further intervention/participation based something that happens post-randomization. It would seem to be a simpler and less biased design to simply give everyone the same intervention regardless of any decisions they make after trial enrollment (ie whether to return to the forest), and also to do the same exact monitoring on everyone. If the authors insist on conditioning the intervention and outcomes on something that happens after randomization, I think they should provide a clearer rationale for why they are doing this and why it should not lead to bias (but acknowledge that it could). The second thing is that the statistical analysis plan is not provided, and the statistical analysis section is quite simplistic and does not indicate what the pre-specified analyses are. Other comments are given below.

P5, line 16: how can a sample size be the number of outcomes? Usually a sample size would be the number of people allocated to each treatment (some of whom will develop the outcome).

P5, line 17: I find this confusing. What does “duration of follow-up and prophylaxis” mean? The duration to me would mean to last follow-up. Study visits can happen periodically for the duration of the study. But the “and prophylaxis” part is the most confusing part of the sentence. I think they mean that the treatment period is 1 month, and that participants will be followed for an additional month off treatment, but it could be stated more clearly.

P5, line 20: what is the primary pre-specified analysis? ITT or per-protocol?

P8, line 25: “after returning from the forest”: any travel to the forest has not really been introduced as a study design element. Will the intervention only be taken while the participants are in the forest? How long will they be in the forest?

P8, line 50: perhaps state how long the course of prophylactic therapy will be so that the reader can more quickly grasp the implications of the different follow-up visits?

P9, line 4: “routes of spread”? Also, is the objective to obtain the data, or to better understand?

P9, line 19: can you provide even a rough estimate of the prevalence or distribution of malaria parasitemia in the study area?

P9, line 43 (and 50): please define rate. Presumably the numerator is PCR positivity. What is the denominator?

P9, line 45 (and 53): wordsmithing, but when it says “proportion of patients with confirmed malaria of any species,” it is a little unclear if the denominator is all patients or instead only those patients with confirmed malaria. If all patients, consider instead “proportion of participants confirmed to have malaria of any species...”

P9, line 56: a “description” is not really an endpoint; what is the metric that will be described

P10 line 5 (and 7 and 10): these are not endpoints. Unless you think that the assigned treatment would change these things? (Though for number 8, you can't change the baseline prevalence)

P10, line 33: I don't have a good sense of how long people stay in the forest. Can a participant be enrolled in the trial multiple times? Or only once? Might be good to specify here since presumably they are making multiple trips to the forest but presumably they cannot be enrolled for trips within the 3-month period of initial enrollment? (I see this is discussed later but it is confusing to read about it up here before knowing what this is)

P10, line 35: will a pregnancy test be administered?

P10, line 42: diarrhea/vomiting over what time period?

P10, line 49: what is a "period"? Does that just mean there are 3 month-long follow-up periods? Or does it mean 3 separate times when the participant goes to the forest? (I see this is discussed later but it is confusing to read about it up here before knowing what this is)

P10, line 45: why not permuted block sizes?

P12, line 9: for how long is the treatment course? I would add that to the box

P12, line 53: how will these minimum 100 individuals be chosen? First 100? Random sample? (I see later it's a random sample, please include that here)

P12, line 8: it seems a little unconventional (and perhaps wrong? I would consult with a trialist about this, and if they think it's OK I would add some rationale about why it is OK) to randomize someone to an intervention, but then to decide whether to administer the intervention conditioned on some post-randomization metric (ie, the decision to give intervention depends on whether they plan to return to the forest, but their decision to return to the forest happens after randomization. Generally that is not allowed in a randomized trial I don't think, though please confirm with others.). Why not just give everyone a 3-month course? Much simpler to explain, and I would think much less prone to bias

P15, line 46: how will the subset be chosen? Random sample? Will the participant carry the GPS device for the entire 3 months of follow-up?

P19, line 40: please define an episode. Does this mean an episode of traveling to the forest? It's a little confusing to me because presumably they might go back to the forest multiple times during the 3 months of the study, but you are only counting it as a single "episode"? Could you just say enrollments or something? Or don't allow a participant to enroll multiple times? Is this really necessary in terms of meeting the sample size goals?

P19, line 54: this is an unconventional way of performing a sample size calculation. Usually one would have an estimate of the incidence in the control group, and then calculate the sample size needed to have 80% power to detect a certain effect size. (I see this is given in the second paragraph. I might reverse the order of the

	paragraphs since most people will be looking for the type of language given in the second paragraph) P19, line 57: please reconcile the 5% estimate with the estimate of 1/person-500 nights given above. I realize the 5% does not take into account the time frame, but where does the 5% come from? P20, line 11: even if a formal statistical analysis plan is provided elsewhere, I think more information should be given here. Specifically: How will the “rate” mentioned as the primary outcome be calculated. What is the pre-specified primary analysis. What are the pre-specified subgroup analyses. How will you account for non-independence of the same participant being enrolled and randomized twice? (I am actually not sure it needs to be accounted for; it could be that the re-randomization takes care of this, but good to reassure the reader and provide some rationale/references in this
--	---

VERSION 1 – AUTHOR RESPONSE

Reviewer reports:

Reviewer: 1

Reviewer Name: Dr Daniel Cooper

Institution and Country: Cambridge University Hospitals, Cambridge, UK; Menzies School of Health Research, Darwin, Australia

The authors present a protocol for a an open-labelled randomised controlled trial to assess the efficacy of artemether-lumefantrine prophylaxis for malaria among forest goers in Cambodia. They present a well-designed and achievable trial that addresses an important public health question in the region. I have no major concerns about the study design or potential of the proposed trial to assess the hypotheses presented. The sample size justification is well considered and based on robust prior studies and data.

I have listed some minor points below, mostly for clarification and ease of understanding of the protocol. The SPIRIT guideline points listed at the end should be specifically addressed and included, as per BMJ Open reviewer guidance and SPIRIT guidelines.

1. The authors state that previous interventions with the same aim have had disappointing uptake in field studies (Pg 4, Line 26). There is no discussion about potential adherence (or lack) to prophylaxis. Why would prophylaxis be better adhered to than previous interventions?

We do not know that adherence to prophylaxis will be better than previous interventions and aim to quantify this as part of the study. We would expect adherence to be higher than the example mentioned of insecticide treated hammock nets. We have added some additional text on page lines 24-25 illustrating why uptake of these has been low in previous studies: “for a variety of reasons including incompatibility with traditional sleep arrangements at home or in the forest”. This limitation does not apply to prophylaxis. A major component of this study is

community engagement to encourage high uptake and adherence to prophylaxis. We have added the following text to emphasise the importance of this: "provided people can be persuaded to take it." And on page 13, lines 22-24 we state: "To maximise adherence to the study medication, the study will be preceded by a period of community sensitisation and engagement including information sessions on the importance of taking all three doses of medication."

2. This is an open-label study. Why are multivitamins being used in the "control" arm? Is there rationale for a benefit of multivitamins in malaria prophylaxis? If this is intended as a "placebo", this should be discussed. If not, and there is no equipoise between the two interventions, this should be more explicit. Addendum: The authors state on page 11 that no components included in the multivitamin have antimalarial activity. The reason for choosing this as a comparator arm therefore needs to be discussed in more detail.

Multivitamins were chosen for the control arm as there was no placebo available from the manufacturer. The multivitamin has no effect on malaria, is safe, acceptable to the community and is easily available at the study site. During the design of the study we consulted with our local partners who understand the remote communities where we work. We were advised that all participants would want receive "something to take". Therefore providing multivitamins to controls made it easier to explain the study in a way that was socially acceptable, and potentially this may have discouraged the sharing of study drugs by participants in the two study arms.

The open label design is sub-optimal when compared to a placebo-control arm in terms of the assessment of non-severe adverse events and other subjective assessments, but is equivalent to placebo for the ascertainment of firm endpoints such as clinical malaria and plasmodium infections detected by PCR. Artemether-lumefantrine is the most widely used artesunate combination therapy (ACT) in the world and its safety and tolerability is already well-described. Allocation to both arms entails taking tablets daily and so to some extent ensures comparability i.e. behaviour among patients in both study arms is as similar as possible.

We have added the following text to page 13, lines 3 to 7: "A multivitamin was chosen because a placebo was not available from the manufacturer for this trial. The multivitamin has no effect on malaria, is safe, acceptable to the community and is easily available at the study site. Providing a medication to all participants makes it easier to explain the study in a way that is socially acceptable, and has potential to discourage the sharing of study drugs by participants in the two study arms."

3. There is no discussion of other Plasmodium species with important transmission within the SouthEast Asian region. In particular, P. knowlesi has key transmission within forest goers and forest dwellers (Grigg et al, Lancet planet Health, 2017) and has been confirmed in clinical cases in Cambodia (Khim et al, EID, 2011).

The primary outcome includes all species of malaria (see page 6, lines 3-5). We have changed "Plasmodium falciparum" on page 4, lines 44 to Plasmodium to reflect this. And added "and will seek to identify all species of human malaria including falciparum, vivax and knowlesi." On page 5, lines 1-2.

4. The authors discuss the ability to detect low density, sub-clinical infection (inferred as a benefit to aid sample size and analysis) Pg 4, Line 49. Is there evidence of benefit to detecting and/or treating low-density subclinical infection? Accepting that this isn't an aim of the study, this should be discussed in terms of extrapolating clinical/public health benefit from the outcomes.

Yes, the goal of prophylaxis is to interrupt or reduce transmission (page 5, lines 1-2) and this includes transmission from both asymptomatic and symptomatic individuals including those with low parasite density. Asymptomatic/low density infections have been shown to be a major source of transmission in Southeast Asia. We have added the following text and an additional reference: "It is important to do so as low density and asymptomatic infections are an important source of malaria transmission in Southeast Asia.[18]" on page 4, lines 44-45.

5. As the primary objective currently reads, it suggests essentially a composite endpoint (PCR positivity OR incidence of confirmed clinical malaria), however the endpoints are listed as coprimary endpoints. It is not currently clear whether this will be analysed as a composite endpoint, or as two separate endpoints. Both are valid, but either way, I think this needs more clarification, and explicit acknowledgement somewhere within the manuscript of a composite endpoint (not necessarily within the main definition) If that is the plan.

We have added the following text on page 20, line 34: "The co-primary endpoints will be analysed as a composite endpoint."

6. It is not clear whether PCR positivity is exclusively within asymptomatic participants. This should be more clear.

Apologies. This PCR is done in all participants including clinical cases. We have added "PCR and" on page 10, line 28.

7. It is not clear to what extent infections/individuals within the two co-primary endpoints will overlap (if not a composite endpoint). This could be acknowledged/discussed.

This is a composite endpoint (see above).

8. The reason for an upper age limit on inclusion should be discussed.

This was a decision based on prior experience and data from the study site. Over 65 year-olds in the study area rarely travel to the forest and are very unlikely to get malaria. They thus fall outside the target group for the intervention. We have added the following text on page 9, lines 27-28 "The upper age limit was chosen as people over 65 years in the study area rarely travel to the forest and are at low risk of malaria."

9. artemether-lumefantrine is considered safe in second and third trimesters of pregnancy, the reason for exclusion of this group should be discussed.

This was a conservative measure to minimize risk. There is a lack of evidence for the safety of artemether-lumefantrine in pregnancy, the stage of pregnancy is often uncertain in this population and the required dosing for prophylaxis in pregnant women is not known.

We have added the following text: "All pregnant women were excluded as a conservative measure to minimize risk because of insufficient evidence for safety of artemether-lumefantrine in the first trimester together with frequent uncertainty about the stage of pregnancy as well as lack of consensus about the required dose in pregnancy." On page 9, lines 28-31.

We believe this exclusion criterion will have affected very few women who might otherwise have been eligible for enrollment. We aim to avoid unnecessary exclusion criteria so that all populations can participate in research, however, in this instance the study is of prophylaxis, not treatment and so a more conservative approach to the risk-benefit of participation is appropriate.

10. Will randomisation be stratified by site?

Randomization was stratified by village but villages will be combined in the analysis. We have added this on page 9, lines 15-16.

11. The authors discuss on page 8, lines 53 – 56 that this is an open-labelled trial and therefore blinding will not be achieved to participants or investigators. However, it is worth discussing that one of the co- primary endpoints is entirely objective (PCR positivity) with blinded laboratory investigations, however the diagnosis of clinical malaria may in theory be subject to bias (however unlikely that is).

We have added the following on page 12, lines 9-10: “The laboratory will be blinded to the study arm of the patient.” And on page 11, lines 24-27: “As clinical malaria at follow-up is one of the co-primary outcomes, and the participants and field staff will not be blinded as to study arm,

there is potential for bias if, for example, people in the AL arm choose not to attend for a malaria test. However, extensive efforts will made through community engagement and individual counselling to advise participants against this.”

12. The rationale for choosing follow-up duration is not discussed.

We have added the following text on page 10, lines 11-12: “28 days was chosen as the upper limit of the time from infection to detectable parasites in the blood.”

13. The authors have provided a brief statistical analysis plan. They have not, however, stated how the co-primary endpoints will be analysed. This should be stated (see further below). Specific SPIRIT guideline points to be clarified (please include each of these points within the body of the text as per SPIRIT guidelines):

Allocation implementation (Item 16 c).

Thanks for the comment. Allocation implementation is described in the section “Randomisation, allocation and blinding”. We have included “allocation” in the subheading to make it clearer.

The person(s) who will enrol/assign participants Data collection methods - retention (Item 18 b).

Thanks for the comment. We have now added some text to address this under “Retention, Discontinuation/ Withdrawal of Participants from the Study”

Strategies to promote participant retention and complete follow up (e.g. phone call reminders, financial compensation).

We have now added some text to address this under “Retention, Discontinuation/ Withdrawal of Participants from the Study”

Statistical methods (Item 20 a)

The main analysis of the primary outcome including the analysis methods to be used for statistical comparisons. The effect measure for the primary outcome (e.g. OR, RR, difference in means).

Significance level and/or intended use of confidence intervals.

*Thanks for the comment. This has now been explicitly stated in the “**Statistical Analyses**”.*

Differences in proportions (risk differences) will be reported.

The following text has been added (page 18, lines 28-46): “The main analysis strategy for the primary outcome will be the intention-to-treat (ITT) principle followed by the per protocol (PP) analysis. Thus, we will first analyse the ITT population in which all participants recruited in the trial will be included in the analysis according to the randomisation arm irrespective of what they actually got. These ITT analyses will be followed by the analysis of the per protocol (PP) population in which participants who did not adhere to the protocol will be excluded from analysis.

The co-primary endpoints will be analysed as a composite endpoint. For 28-day PCR Plasmodium positivity and parasite positive clinical episode rate analysis, each arm will be summarised using

crude proportions and binomial exact 95% confidence intervals. The risk differences in Plasmodium positivity between AL versus Multivitamin will be reported along with the corresponding 95% confidence intervals. Robust standard errors will be used to handle multiple episodes. These analyses will be complemented by the use of the crude Kaplan-Meier estimates of cumulative PCR Plasmodium positivity and parasite positive clinical episode probabilities as recommended by WHO. The incidence of confirmed clinical malaria between day 0 to day 28 analysis will be modeled using the mixed effects Poisson regression model to obtain incidence rate ratios (IRR) comparing AL versus Multivitamin arms. The mixed effects models will take into account the correlation of multiple episodes from the same participant. Tests of significance will be performed at 5% significance level. Analysis of all endpoints will be described in detail in a Statistical Analysis Plan. A brief overview is given below.”

Population analysed (Item 20 c)

Clear indication of which participants will be included in the main analysis. Simply stating intention to treat or per-protocol without further specification is not sufficient.

*Thanks for the comment. This has now been explicitly stated in the “**Statistical Analyses**” (as above). Both the ITT and the per protocol analyses have been expanded.*

How missing data will be handled (or a description of why missing data is unlikely)

Thanks for the comment. A section dedicated to handling missing data has been added to the text. Details of how missing data will be handled for the primary outcome has been explained.

Reviewer: 2

Reviewer Name: Chaturaka Rodrigo

Institution and Country: UNSW Sydney, Australia

I have read this protocol with interest, which is well-written and addresses an important research question. However, I have the following concerns regarding some aspects of the protocol

1. While the randomisation process for individual recruits are explained, these individuals will be selected from particular villages and the random selection of villages is not explained

Villages are not randomly selected. All villages in the study area with reported malaria cases are included.

2. If this is an open label trial, what is the advantage of giving multivitamin tablets to control group compared to no treatment?

Multivitamins were chosen for the control arm as there was no placebo available from the manufacturer. The multivitamin has no effect on malaria, is safe, acceptable to the community and is easily available at the study site. During the design of the study we consulted with our local partners who understand the remote communities where we work. We were advised that

all study participants would want to receive "something to take". This made it easier to explain the study in a way that was socially acceptable, and potentially this may have discouraged the sharing of study drugs by participants in the two study arms.

The open label design is sub-optimal when compared to a placebo-control arm in terms of the assessment of adverse events and subjective measures, but is equivalent to placebo for the ascertainment of firm endpoints such as clinical malaria and plasmodium infections detected by PCR. Artemether-lumefantrine is the most widely used artesunate combination therapy (ACT) in the world and its safety and tolerability is already well-described. Allocation to both arms entails taking tablets daily and so to some extent ensures comparability i.e. behaviour among patients in both study arms is as similar as possible.

We have added the following text to page 13, lines 3 to7: "A multivitamin was chosen because a placebo was not available from the manufacturer for this trial. The multivitamin has no effect on malaria, is safe, acceptable to the community and is easily available at the study site. Providing a medication to all participants makes it easier to explain the study in a way that is socially acceptable, and has potential to discourage the sharing of study drugs by participants in the two study arms."

3. One of the objectives refer to a "mathematical modelling" with no details of how this will be done in the protocol

We have added details about this on page 19, lines 32-36: "The impact of the ACT artemether-lumefantrine as prophylaxis for forest goers on overall malaria transmission will be quantified using mathematical modelling. For this we will develop a population dynamic village level individual-based model of malaria transmission and treatment parameterised with published data, results from the analysis of data from the trial and fitted to surveillance data from the study area."

4. There are several objectives (examining high risk behaviours) that may be beyond the scope of the drug efficacy trial protocol (while I agree these are important for prevention of malaria, the purpose of including these details in this protocol is unclear)

The purpose of these objectives is to gain a better understanding of the study population. This is because this trial is planned to inform government decision making about future roll-out of prophylaxis as an intervention.

5. There is a risk of including patients with subclinical malaria infection at baseline and saving a sample for later PCR does not solve this problem. Alternatively you may restrict your analysis to clinical malaria

This was deliberate and is part of the reason why we begin the course of prophylaxis with a full 3 days of treatment to clear any subclinical infection. It is not possible to get results from the PCR quickly enough to make a decision about enrolment. Restricting the analysis to clinical malaria would mean we would need a much larger sample size as a substantial proportion of infections are asymptomatic. It would also mean we could not fully quantify the impact of prophylaxis on transmission and consequent potential contribution to malaria elimination as asymptomatic infections contribute greatly to transmission in this region. We have added the following: "It is

important to do so as low density and asymptomatic infections are an important source of malaria transmission in Southeast Asia.[18]" on page 4, lines 45-46.

6. The procedure for handling missing data is not clear apart from that data collected up to the last point of enrolment will be included for dropouts. Will it be a intention to treat analysis?

Thanks for the comment. A section dedicated to handling missing data has been added to the text. Details of how missing data will be handled for the primary outcome has been explained. The intention to treat has also been explained in the revised text.

7. I can foresee some serious ethical concerns with some types of data collected (GPS data on participants) and this protocol should ideally be approved by an ethics review committee and the editor may decide if it should be a pre-requisite for publication

This protocol has been approved by ethical committees of Oxford University and the Cambodian government and the trial is already underway.

8. Testing for genetic markers is vague (e.g. Kelch13 and other genetic markers...). Please specify what you are specifically looking for

We are looking for an increase in markers of antimalarial drug resistance. As this will be done by sequencing, we will be able to detect any known markers. We deliberately kept this open as work is ongoing to identify new markers and we did not want to restrict the protocol to markers known at the time of preparing the study protocol.

9. If a patient had clinical malaria at baseline, is he/she eligible to be recruited after treatment and eradication of infection?

Yes. We have added the text "Such individuals may be enrolled later following recovery provided they meet the inclusion and exclusion criteria." To page 9, lines 9-10.

10. If the same person is re-enrolled (as mentioned in the protocol), how will this be addressed statistically as it is incorrect to use the same statistical test when some observations are within host and others are between-host. Several other comments (including the ones listed above) are also given as annotations in the attached PDF

Thanks for the comment. We agree with the reviewer. As detailed in the analysis section. Methods that takes intracluster correlations into account will be used to adjust the analyses for re-enrolment. We have detailed this in the "Statistical Analysis" section. We will finalise the statistical analytical plan prior to locking the database.

Reviewer: 3

Reviewer Name: Jeremy Keenan

Institution and Country: UCSF, USA

This is a trial protocol of a randomized trial comparing antimalarial prophylaxis with artemether lumefantrine vs a multivitamin control among forest-goers in Cambodia. My main comments are that I do not completely understand the rationale for conditioning further intervention/participation based something that happens post-randomization. It would seem to be a simpler and less biased design to simply give everyone the same intervention regardless of any decisions they make after trial enrollment (ie whether to return to the forest), and also to do the same exact monitoring on everyone. If the authors insist on conditioning the intervention and outcomes on something that happens after randomization, I think they should provide a clearer rationale for why they are doing this and why it should not lead to bias (but acknowledge that it could).

Apologies if the rationale for this was not clear in the protocol. Patients are randomized at the beginning and remain in the same study arm throughout follow-up. We chose to have different numbers of episodes per individual for practical reasons based on baseline information that habits for repeated forest visits vary widely between individuals. With some people visiting the forest repeatedly over several months, and others only visiting once during the same period. We wanted to maximise study power for the limited resources and budget available by not continuing follow-up of people who were no longer visiting the forest and therefore were not at risk of catching forest malaria as this would be a lot of effort to detect very few or no additional infections.

An additional factor that guided this design was feedback from the study communities that people who no longer visited the forest were likely to stop returning for follow-up. In order to maximise power, we therefore decided on a strategy which would include only 28 periods with significant risk of malaria transmission.

This was also a trial designed to guide decisions about implementation of prophylaxis as an intervention by the Cambodian government. It was felt that it would not be appropriate to administer prophylaxis to people who were no longer at significant risk of infection.

We acknowledge that if not appropriately accounted for in the analysis, having variable episodes per individual could lead to bias. However, we have planned the analysis to minimize this, as outlined above.

The second thing is that the statistical analysis plan is not provided, and the statistical analysis section is quite simplistic and does not indicate what the pre-specified analyses are. Other comments are given below.

We have greatly expanded the sections on statistical analysis (page 18, lines 28 to 46 and Handling of Missing data (page 19, lines 17-24). We will finalise the statistical analytical plan prior to locking the database.

P5, line 16: how can a sample size be the number of outcomes? Usually a sample size would be the number of people allocated to each treatment (some of whom will develop the outcome).

This was a deliberate choice as the intervention is only relevant for people who visit the forest. Therefore we did not want to continue to follow people up if they did not do this. Behaviours

about travel to the forest vary widely between individuals and were not controlled in the trial. We will account for the number of episodes individuals participate in the study in the analysis.

P5, line 17: I find this confusing. What does “duration of follow-up and prophylaxis” mean? The duration

to me would mean to last follow-up. Study visits can happen periodically for the duration of the study. But the “and prophylaxis” part is the most confusing part of the sentence. I think they mean that the treatment period is 1 month, and that participants will be followed for an additional month off treatment, but it could be stated more clearly.

The duration of follow-up varies between 1 and 3 28-day episodes depending on how long people continue to visit the forest. The duration of prophylaxis was 28 days longer than this to clear any remaining parasites that may appear after leaving the forest. This is a standard part of antimalarial chemoprophylaxis e.g. in travelers.

P5, line 20: what is the primary pre-specified analysis? ITT or per-protocol?

Thanks for the comment. This has now been well detailed in the “Statistics Analysis” section. The main strategy is ITT.

P8, line 25: “after returning from the forest”: any travel to the forest has not really been introduced as a study design element. Will the intervention only be taken while the participants are in the forest? How long will they be in the forest?

The study was designed to assess the real-world impact of antimalarial prophylaxis in forest goers. We did not intervene to influence participant’s travel plans to the forest and for 4 weeks after they left (see above) so they were free to choose the number of periods of participation. Patterns of forest visits vary widely. Therefore we included any travel to the forest where they stayed overnight, the latter being because most malaria transmission happens at this time.

P8, line 50: perhaps state how long the course of prophylactic therapy will be so that the reader can more quickly grasp the implications of the different follow-up visits?

Thanks. We have added the following text to clarify: “Follow-up will be for 1-3 periods of 28 days each depending on whether the participant continues to visit the forest.”

P9, line 4: “routes of spread”? Also, is the objective to obtain the data, or to better understand?

Sorry we did not understand the first part of this comment. For the second part, objective 6 is both to collect data, as stated. This is similar to objective 4.

P9, line 19: can you provide even a rough estimate of the prevalence or distribution of malaria parasitemia in the study area?

Sorry we had no prior data on prevalence in the study area. But we knew from the village malaria workers which villages had cases reported. From incidence data reported to the government, transmission intensity was known to be low and varying from year to year. We have added this text on page 6, line 32: “Malaria transmission in this area is generally low but varying over time.”

P9, line 43 (and 50): please define rate. Presumably the numerator is PCR positivity. What is the denominator?

This is the proportion of blood samples which were PCR positive. We have added this on page 7, line 22. We will finalise the statistical analytical plan prior to locking the database.

P9, line 45 (and 53): wordsmithing, but when it says “proportion of patients with confirmed malaria of any species,” it is a little unclear if the denominator is all patients or instead only those patients with

confirmed malaria. If all patients, consider instead “proportion of participants confirmed to have malaria of any species...”

Apologies, we think the reviewer may have misread this Endpoint. We used proportion of participants, not patients. Therefore this is the proportion of people participating in the study which had confirmed clinical malaria.

P9, line 56: a “description” is not really an endpoint; what is the metric that will be described

The term “description” was chosen as it was not clear at the time of writing the protocol what surveillance data would be available to us therefore we could not specify in detail. However, at a minimum it will include the number of cases by village, and demographics of those cases. We have added this in the text (page 7, line 23).

P10 line 5 (and 7 and 10): these are not endpoints. Unless you think that the assigned treatment would

change these things? (Though for number 8, you can't change the baseline prevalence)

We respectfully disagree. This is a trial with embedded epidemiological studies and endpoints which are independent of the trial intervention. This is common practice in trials of this type.

P10, line 33: I don't have a good sense of how long people stay in the forest. Can a participant be enrolled in the trial multiple times? Or only once? Might be good to specify here since presumably they are making multiple trips to the forest but presumably they cannot be enrolled for trips within the 3- month period of initial enrollment? (I see this is discussed later but it is confusing to read about it up here before knowing what this is)

The time people stay in the forest varies widely from a day to several weeks. We explicitly state that participants may be enrolled a maximum of 3 times on page 13, line 16. We have added the text "As detailed below, participants may be enrolled a maximum of 3 times during the study period." On page 8, lines 34-35 to emphasise this as suggested by the reviewer.

P10, line 35: will a pregnancy test be administered?

No. Only known pregnancies are excluded as stated in the exclusion criteria.

P10, line 42: diarrhea/vomiting over what time period?

On the day of screening. We have added this to the text.

P10, line 49: what is a "period"? Does that just mean there are 3 month-long follow-up periods? Or does it mean 3 separate times when the participant goes to the forest? (I see this is discussed later but it is confusing to read about it up here before knowing what this is)

It means 3 consecutive periods of 28 days. This is explained earlier on page 6, lines 36-37 and we have added "consecutive" to clarify that it is not 3 separate times.

P10, line 45: why not permuted block sizes?

Thanks for the comment. We have made it explicit that we used permuted blocks

P12, line 9: for how long is the treatment course? I would add that to the box

The duration of the course of prophylaxis varies with the duration of follow-up as described in detail on page 10, lines 31 to 40 and page 11 lines 1-3. The box summarises only the frequency of dosing.

P12, line 53: how will these minimum 100 individuals be chosen? First 100? Random sample? (I see later

it's a random sample, please include that here)

We have added the suggested text.

P12, line 8: it seems a little unconventional (and perhaps wrong? I would consult with a trialist about

this, and if they think it's OK I would add some rationale about why it is OK) to randomize someone to an intervention, but then to decide whether to administer the intervention conditioned on some post-randomization metric (ie, the decision to give intervention depends on whether they plan to return to the forest, but their decision to return to the forest happens after randomization. Generally that is not allowed in a randomized trial I don't think, though please confirm with others.). Why not just give everyone a 3-month course? Much simpler to explain, and I would think much less prone to bias

We apologise if this was not clear. Participants continued on the same intervention throughout follow-up and there was no decision to administer or not following randomization. If the participant declared at follow-up that they would not visit the forest again in the next 28 days, they then completed the study.

As outlined above, the decision to have 1, 2 or 3 periods of follow-up was a pragmatic one to maximise study power whilst minimizing use of resources. This is because individuals who no longer visited the forest were unlikely to catch malaria and would not contribute an endpoint during those periods. In addition, it was felt that adherence to prophylaxis would decrease in people who stopped visiting the forest and therefore had low risk of malaria. And the trial was designed to provide evidence to the Cambodian government for possible implementation of prophylaxis as an intervention and it was not felt appropriate to give prophylaxis to people who were no longer at risk.

The study team are themselves experienced trialists and the protocol has also undergone internal scientific review with senior staff with extensive experience of clinical malaria trials who were not involved in the study.

P15, line 46: how will the subset be chosen? Random sample? Will the participant carry the GPS device for the entire 3 months of follow-up?

The selection of the subset to be given GPS devices is described on page 13, lines 42-46 and page 14 lines 1-2. We have added the word "consecutive" to clarify that consecutive unselected participants will be given the devices until there are none left out of the 50 available. Once the

devices are returned on completion of follow-up they will then be given to other individuals. They will be carried for the entire follow-up period: we have added " , being returned upon completion of follow-up for that individual" to clarify this.

P19, line 40: please define an episode. Does this mean an episode of traveling to the forest? It's a little confusing to me because presumably they might go back to the forest multiple times during the 3 months of the study, but you are only counting it as a single "episode"? Could you just say enrollments or something? Or don't allow a participant to enroll multiple times? Is this really necessary in terms of meeting the sample size goals?

Apologies for any confusion. We have added the following text to clarify this: "An episode is defined as a follow-up period of 28 days with each enrolled individual contributing 1, 2 or 3 episodes." As described earlier, participants go back to the forest whenever they choose and this varies widely between individuals and is not possible to control. We use episodes rather than enrolments for reasons outlined above as each enrolled person can contribute multiple episodes. This design was indeed necessary to meet the sample size goals.

P19, line 54: this is an unconventional way of performing a sample size calculation. Usually one would have an estimate of the incidence in the control group, and then calculate the sample size needed to have 80% power to detect a certain effect size. (I see this is given in the second paragraph. I might reverse the order of the paragraphs since most people will be looking for the type of language given in the second paragraph)

Thank you. We have chosen to delete the text referred to in order to maximise clarity.

P19, line 57: please reconcile the 5% estimate with the estimate of 1/person-500 nights given above. I realize the 5% does not take into account the time frame, but where does the 5% come from?

We have removed the text referring to person-nights. The 5% is an estimate of the risk of infection per 28 day episode.

P20, line 11: even if a formal statistical analysis plan is provided elsewhere, I think more information should be given here. Specifically: How will the “rate” mentioned as the primary outcome be calculated. What is the pre-specified primary analysis. What are the pre-specified subgroup analyses. How will you account for non-independence of the same participant being enrolled and randomized twice? (I am actually not sure it needs to be accounted for; it could be that the re-randomization takes care of this, but good to reassure the reader and provide some rationale/references in this case).

Thanks for the comment. We have now included more information from the analysis plan. This has been added to text in the “Statistics Analysis” section.

VERSION 2 – REVIEW

REVIEWER	Chaturaka Rodrigo UNSW Sydney, Australia
REVIEW RETURNED	20-Mar-2021

GENERAL COMMENTS	My concerns ave been satisfactorily addressed by the authors. I have no further comments
--

REVIEWER	Jeremy Keenan UCSF, USA
REVIEW RETURNED	25-Mar-2021

GENERAL COMMENTS	Thank you for making revisions. I do not see a “response to reviewer’s comments” document. Several of my prior comments were not addressed. I don’t see any changes to the abstract. I will simply copy and paste my earlier comments, none of which have been addressed: --“The sample size is 2200 patient- episodes of duration 1 month in each arm.” □ how can a sample size be the number of outcomes? Usually a sample size would be the number of people allocated to each treatment (some of whom will develop the outcome). --“The duration of follow-up and prophylaxis for each participant is 1, 2 or 3 consecutive 28 day periods, followed by a further 28 days of post-exposure prophylaxis, depending on whether they continue to
---

	visit to the forest.” □ I find this confusing. What does “duration of follow-up and prophylaxis” mean? The duration to me would mean to last follow-up. Study visits can happen periodically for the duration of the study. But the “and prophylaxis” part is the most confusing part of the sentence. I think they mean that the treatment period is 1 month, and that participants will be followed for an additional month off treatment, but it could be stated more clearly. --“Analysis will be done both by intention-to-treat and per-protocol.” □ what is the primary pre-specified analysis? ITT or per-protocol? Exclusion criteria: as commented in the prior review: is this pregnancy by self-report? Or will a pregnancy test be done for all women (or offered) Study drug administration: as commented in the prior review, please provide more detail in the the “Study drug administration” box. Specifically, provide how many times per day for each drug, and the duration of treatment for each drug. “Follow up”, “...any time from 28 to 35 days after commencing prophylaxis” □ most readers will be unfamiliar with the normal duration of stay in the forest. Do forest workers typically stay a single day? A week? A month? What if they are in the forest for 3 months and cannot come to your follow-up visit? (I see this is later addressed, that they should present at the first opportunity... but I might still give some context about typical durations in the forest for the reader) “Individuals who have been enrolled in the study may be enrolled into the study up to two more times...” □ It would be good to give some rationale. Why even allow this? Is it because it will be difficult to enroll? And then why a maximum of three enrollments? “co-primary endpoints will be analysed as a composite endpoint.” □ Please define what this composite means. It is parasitemia and/or clinical malaria? “co-primary endpoints will be analysed as a composite endpoint.” □ perhaps semantics, but I think of co-primary outcomes and a composite outcome as 2 different concepts. My sense is that there is a single primary outcome here, and that the outcome is a composite outcome of parasitemia and/or clinical malaria, and that only a single analysis will be done. “Co-primary outcomes” suggests that 2 separate analyses will be done, one for parasitemia and one for clinical malaria. I would change the “co-primary” outcome language in the paper in favor of the “composite outcome” language if that is indeed what is being done.
--	--

VERSION 2 – AUTHOR RESPONSE

Reviewer: 3

Dr. Jeremy Keenan, University of California, San Francisco

Comments to the Author:

Thank you for making revisions. I do not see a “response to reviewer’s comments” document.

Several of my prior comments were not addressed.

I don't see any changes to the abstract. I will simply copy and paste my earlier comments, none of which have been addressed:

We really appreciate all your feedback and suggestions on our protocol. Our apologies that you did not receive the "response to reviewer's comments" file. This was uploaded to the journal submission site on 24th February 2021 together with the revised manuscript with and without tracked changes.

For your information, we have again uploaded our original response to reviewer's comments document (First Response to reviewers for PAL Trial Manuscript ID bmjopen_24Feb21.docx) and hope this satisfactorily addresses your earlier comments. In particular we greatly expanded the sections on statistical analysis (page 18, lines 28 to 46 and Handling of Missing data (page 18, line 41 – page 19 line 41).

In addition, we have prepared a new updated response to your latest comments with the relevant parts of our previous response inserted and some further changes and clarifications in underlined text which you will find below.

--"The sample size is 2200 patient- episodes of duration 1 month in each arm." ◇ how can a sample size be the number of outcomes? Usually a sample size would be the number of people allocated to each treatment (some of whom will develop the outcome).

This was a deliberate choice as the intervention is only relevant for people who visit the forest. Therefore we did not want to continue to follow people up if they did not do this as they would not be continuing the intervention or having outcomes measured. Behaviours about travel to the forest vary widely between individuals and were not controlled in the trial. We will account for the number of episodes individuals who participate in the study in the analysis.

--"The duration of follow-up and prophylaxis for each participant is 1, 2 or 3 consecutive 28 day periods, followed by a further 28 days of post-exposure prophylaxis, depending on whether they continue to visit to the forest." ◇ I find this confusing. What does "duration of follow-up and prophylaxis" mean? The duration to me would mean to last follow-up. Study visits can happen periodically for the duration of the study. But the "and prophylaxis" part is the most confusing part of the sentence. I think they mean that the treatment period is 1 month, and that participants will be followed for an additional month off treatment, but it could be stated more clearly.

Sorry if this was not clear. It actually is the converse – the duration of prophylaxis is 28 days longer than the duration of follow-up. The duration of follow-up varies between 1 and 3 28-day episodes depending on how long people continue to visit the forest. The duration of prophylaxis is 28 days longer than this to clear any remaining parasites that may appear after leaving the forest. This is a standard part of antimalarial chemoprophylaxis e.g. in travelers. This is mentioned in the abstract on page 2, line 15 and page 11, lines 11 and 14-15.

--“Analysis will be done both by intention-to-treat and per-protocol.” ◇ what is the primary pre-specified analysis? ITT or per-protocol?

Thanks for the comment. This has now been detailed in the “Statistics Analysis” section. The main strategy is ITT.

Exclusion criteria: as commented in the prior review: is this pregnancy by self-report? Or will a pregnancy test be done for all women (or offered)

Only known pregnancies are excluded as stated in the exclusion criteria. This is therefore by self-report. We have added this on page 8, lines 38.

Study drug administration: as commented in the prior review, please provide more detail in the the “Study drug administration” box. Specifically, provide how many times per day for each drug, and the duration of treatment for each drug.

Thank you for this new comment about daily dosing frequency. We have now added “once daily” and “twice daily” to the box, as suggested.

The duration of the course of prophylaxis varies with the duration of follow-up as described in detail on page 10, lines 13-15 and page 11 lines 3-15 and this is difficult to summarise in the box. The box was intended only to summarise the frequency of dosing. We hope you agree.

“Follow up”, “...any time from 28 to 35 days after commencing prophylaxis” ◇ most readers will be unfamiliar with the normal duration of stay in the forest. Do forest workers typically stay a single day? A week? A month?

Thank you for this new comment suggesting to add detail about the duration of forest trips. We have added the following text: “Ongoing studies in the area found the duration of forest visits varied from a day to several weeks, with very few people being away for more than 28 days.” on page 10, lines 19-21 to clarify this further.

You had a previous similar comment about the number of trips which we had addressed with the following: The time people stay in the forest varies widely from a day to several weeks. We state that participants may be enrolled a maximum of 3 times on page 11, lines 16-18. We have added the text “As detailed below, participants may be enrolled a maximum of 3 times during the study period.” on page 8, lines 36-37 to emphasise this as suggested by the reviewer.

What if they are in the forest for 3 months and cannot come to your follow-up visit? (I see this is later addressed, that they should present at the first opportunity... but I might still give some context about typical durations in the forest for the reader)

Thank you for this new comment asking about inability to attend for follow-up. People who cannot return for follow-up meet the exclusion criterion “4. Willingness and ability of the participants to comply with the study protocol for the duration of the study.” and would not

therefore be enrolled. We have added “people that cannot return for follow-up will not be enrolled.” on page 8, line 36 to clarify this further. For participants who do not return for follow-up at the scheduled time after enrollment this is addressed on Page 10, lines 37-38: “If the person cannot be followed up within the scheduled period, e.g. because they do not return from the forest in time, then they will be followed up at the first opportunity and this will be recorded in the CRF.”

We have added some context about typical durations as described for the previous comment.

“Individuals who have been enrolled in the study may be enrolled into the study up to two more times...” ◊ It would be good to give some rationale. Why even allow this? Is it because it will be difficult to enroll? And then why a maximum of three enrollments?

Thank you for this new comment asking for rationale. We added the following text on page 11, lines 20-24. “The rationale for this re-enrolment was that malaria transmission and forest travel are seasonal at this location and this allowed detection of malaria positive episodes in people who continue to visit the forest throughout the year whilst minimizing the period of follow-up for the majority of people who visit the forest only during a particular season; in addition, it allowed a wash-out period between episodes of taking prophylaxis.”

“co-primary endpoints will be analysed as a composite endpoint.” ◊ Please define what this composite means. Is it parasitemia and/or clinical malaria?

We have added “and/or” on page 9, line 5.

“co-primary endpoints will be analysed as a composite endpoint.” ◊ perhaps semantics, but I think of co-primary outcomes and a composite outcome as 2 different concepts. My sense is that there is a single primary outcome here, and that the outcome is a composite outcome of parasitemia and/or clinical malaria, and that only a single analysis will be done. “Co-primary outcomes” suggests that 2 separate analyses will be done, one for parasitemia and one for clinical malaria. I would change the “co-primary” outcome language in the paper in favor of the “composite outcome” language if that is indeed what is being done.

Thank you for this new comment asking us to change the terminology. We have changed co-primary to composite primary throughout as suggested.

VERSION 3 – REVIEW

REVIEWER	Jeremy Keenan UCSF, USA
REVIEW RETURNED	15-Jun-2021
GENERAL COMMENTS	The authors have addressed all comments.